# Navigating the shots: Parental willingness to immunize their children with COVID-19 vaccines in Saudi Arabia explored through a systematic review and meta-analysis

**Moustafa Abdelaal Hegazi**[1,2]*, **Mohamed Hesham Sayed**[1,3], **Nadeem Shafique Butt**[4], **Turki Saad Alahmadi**[1,5], **Nadeem Alam Zubairi**[1], **Wesam Abdelaziz Elson**[6]

1 Faculty of Medicine in Rabigh, Department of Pediatrics, King Abdulaziz University, Jeddah, Saudi Arabia, 2 Faculty of Medicine, Department of Pediatrics, Mansoura University Children's Hospital, Mansoura, Egypt, 3 Faculty of Medicine, Department of Pediatrics, Cairo University, Cairo, Egypt, 4 Faculty of Medicine in Rabigh, Department of Family and Community Medicine, King Abdulaziz University, Jeddah, Saudi Arabia, 5 Department of Pediatrics, King Abdulaziz University Hospital, Jeddah, Saudi Arabia, 6 Department of Obstetrics and Gynecology, Nahdi Care Clinics, Jeddah, Saudi Arabia

* mhegazi712003@yahoo.co.uk, mahhassan@kau.edu.sa

**Data Availability Statement:** All relevant data are within the manuscript and its Supporting Information files.

## Abstract

### Introduction

Although COVID-19 vaccines have been recommended for children and adolescents since 2021, suboptimal vaccination uptake has been documented. No previous systematic review/meta-analysis (SRMA) investigated parents' willingness to administer COVID-19 vaccines for their children in Saudi Arabia. Accordingly, this SRMA aimed to estimate parents' willingness to immunize their children with COVID-19 vaccines in Saudi Arabia and to identify reasons and determinants influencing parents' decisions.

### Methods

This SRMA adhered to the PRISMA guidelines and the protocol was registered on PROS-PERO (ID: CRD42023492760). An extensive systematic search was performed across electronic databases including Pub Med, Pub Med Central, ISI Web of science, Web of Science Core Collection, Medline, KCI-Korean Journal Database, ProQuest, and SciELO, to identify relevant studies published from January 1, 2020 to October 30, 2023. A random-effects model was utilized to estimate the pooled effects considering the expected variability across studies. Heterogeneity, risk of bias, publication bias and quality of studies were considered and evaluated by relevant appropriate tests to ensure robust results.

### Results

Twenty-five studies with 30,844 parents were included. The overall pooled rate of parents who intended to immunize their children with COVID-19 vaccines was 48.0% (95% CI: 41.0–54.0%) with high heterogeneity ($I^2$ = 99.42%). The main reason for parents to vaccinate children was to protect child, family and community from COVID-19. Perceived

**Funding:** The author(s) received no specific funding for this work.

**Competing interests:** The authors have declared that no competing interests exist

efficacy/safety of vaccines were the most significant determinants associated with parents' willingness to vaccinate children.

## Conclusion

This was the first SRMA from Saudi Arabia which emphasized the priority to focus on vaccine-related factors as main/key strategy of COVID-19 vaccines' drivers to convince parents in a logical way based on accurate cumulative and emerging scientific data about efficacy and safety of COVID-19 vaccines to optimize their uptake by children/adolescents. This SRMA can provide valuable insights for development of evidence-based policies to improve parental willingness to vaccinate children, which is crucial for controlling SARS-CoV-2 spread and promoting herd immunity in the community particularly if the virus continues to pose a major threat.

## Introduction

COVID-19 (coronavirus disease-2019), caused by the SARS-CoV-2 virus (severe acute respiratory syndrome coronavirus 2), is a highly contagious illness leading to severe acute respiratory syndrome. It has caused over 7 million deaths globally by 7[th] of June, 2024 and has been considered the most significant health crisis since the 1918 influenza epidemic [1]. After the initial cases were reported in Wuhan, China, in December 2019, the virus quickly spread worldwide. The World Health Organization declared it a global pandemic on March 11, 2020 [2].

In 2020, COVID-19 became the third leading cause of death in the United States, with over 375,000 fatalities, ranking behind heart disease and cancer. Although all people are susceptible to get COVID-19 infection, the Centers for Disease Control and Prevention (CDC) highlights that older adults and individuals with comorbidities are at greater risk for serious illness [3]. According to the Saudi ministry of health, 358,713 COVID-19 cases and 5965 deaths were reported in 2020 [4]. Saudi studies detected that older age, male gender, and comorbidities are risk factors that enhance COVID-19 mortality [5,6].

Children are generally less affected by COVID-19 than adults, but severe cases can occur. A recent global systematic review and meta-analysis (SRMA) of SARS-CoV-2 in children and adolescents found a positivity rate of 9.30%, with most cases being mild. However, 20.70% of pediatric infections were hospitalized, 7.19% needed oxygen support, 4.26% required intensive care, 2.92% needed assisted ventilation, with case fatality rate of 0.87% [7]. Multisystem inflammatory syndrome in children (MIS-C) posed a serious threat of critical illness in hospitalized children. According to the International Kawasaki Disease Registry, despite the clinical severity of MIS-C becoming milder over time, severe cases remained prevalent. During the Omicron period, 23% of MIS-C patients presented with shock, and 37% were admitted to intensive care [8].

Studies on children's role in transmitting COVID-19 within households have shown mixed results. Some research suggests that children are as likely or more likely than adults to spread the virus, while other studies find such transmission to be rare. These discrepancies are due to variations in community infection rates, detection methods, sample collection timing, and adherence to home infection control measures [9,10].

Currently, no medication has been proven effective against SARS-CoV-2, and the main strategy remains prevention. Key preventive measures include patient isolation, rigorous infection control, frequent hand washing, and avoiding touching the face after contact with potentially contaminated surfaces [11].

The remarkable success of routine childhood vaccinations in disease prevention, directed the attention for the importance of global implementation of childhood immunisation programmes to prevent direct sequelae of COVID-19 especially, COVID-19-related hospitalizations, severe complications like MIS-C, long-COVID symptoms, and to mitigate the indirect effects of long quarantine, school closures and other lockdown measures on children's education, behavioural, social development and mental well-being [12].

Since 2021, children have received COVID-19 mRNA vaccines. A SRMA of 17 studies confirmed the efficacy of mRNA vaccine in reducing COVID-19 infection, hospitalizations, and MIS-C among children. The vaccine has been found safe, with a very low incidence of myocarditis, affecting only 1.8 per million recipients. This review also highlighted that mRNA vaccines consistently prevented severe disease during the delta and omicron waves, with minimal risk of myocarditis [13].

Although COVID-19 morbidity and mortality rates are significantly lower in children compared to adults, the risk of serious illness remains notable, even among previously healthy children. However, the response to mRNA vaccines among children aged 5 to 11 has been lukewarm. The estimated worldwide pooled proportion of parents intending to administer COVID-19 vaccines for their children was moderate at 57% in one SRMA [11], and 60.1%, with acceptance varying widely across studies from 25.6% to 92.2% in another SRMA [15]. This lower intention may stem from perceptions that COVID-19 is less severe in children and concerns about vaccine safety in this age group and complicated by the tendency of children to be frequently asymptomatic carriers [14,15]. A recent systematic review exploring parental barriers and facilitators in Western countries identified that the primary barriers were concerns about side effects and distrust in institutions [16]. Furthermore, investigating the predictors of COVID-19 vaccine decision-making among parents of children aged 5–11 in the UK reflected barriers pertaining to time constraints, workday loss, concerns about vaccine induced immunity duration and potential vaccination side effects [17]. The findings from a metropolitan area of the United States indicated that significantly less hesitant parents reported willingness vaccinate their children with a safe, effective COVID-19 vaccine if it were available compared to non-hesitant parents and older mothers with two or more children are more likely to be vaccine hesitant [18].

Moreover, important information that should be conveyed to parents, includes the likely persistence of SARS-CoV-2 for years, the virus can remain dangerous, millions of children born annually will remain susceptible to the virus, the virus may cause severe illness across all age groups, severe COVID-19 has been effectively prevented by mRNA vaccines in over 10 million children aged 5 to 11, and myocarditis is an exceptionally rare side effect of these vaccines in young children [19].

In Saudi Arabia, one recent Saudi study identified that the willingness of parents to vaccinate their children against COVID-19 was 61%. The main reason for taking was "Protect the child" and the main reason for refusing was "Side effects/safety concerns" [20]. In another Saudi study, false religious beliefs were found to significantly influence the opposing attitude towards vaccinating children against COVID-19 [21].

To our knowledge, no previous SRMA investigated parents' willingness to administer COVID-19 vaccines for their children in Saudi Arabia which has religious and cultural characteristics different from other parts of the world. Accordingly, this SRMA aimed to estimate parents' willingness to immunize their children with COVID-19 vaccines in Saudi Arabia and to identify the reasons and determinants influencing parents' decisions. Thus, this research is crucial for health authorities to develop targeted public health interventions to tackle any perceived barriers and utilise any facilitators to enhance vaccination rates and achieving significant control over the propagation of SARS-CoV-2 in the community.

## Methods

This SRMA adhered to the PRISMA guidelines and the protocol was registered on PROSPERO (Date: December 13, 2023, Reference No: CRD42023492760).

The SRMA utilized the population, intervention, comparison, and outcome (PICO) framework to formulate the review questions/outcome. The primary questions addressed were:

1. What is the rate of parents willing to immunize their children with COVID-19 vaccines in Saudi Arabia?

2. What are the reasons of parents to immunize their children with COVID-19 vaccines?

Secondary or additional questions; what factors/determinants/predictors are associated with parents' willingness to immunize their children with COVID-19 vaccines?

Extensive systematic searching was performed across several electronic databases including Pub Med, Pub Med Central, ISI Web of science, Web of Science Core Collection, Medline, KCI-Korean Journal Database, ProQuest, and SciELO. References of all included articles were reviewed to identify additional relevant studies.

The Medical Search Headings (MeSH) terms and keywords were used. The search terms included (parent OR caregiver OR father OR mother) AND (willingness OR adherence OR acceptance OR intention OR compliance) AND (refusal OR hesitancy OR reject OR object) AND (reasons OR facilitators) AND (barriers OR obstacles) AND (predictors OR factors) AND (related to OR associated with) AND (COVID-19 OR Coronavirus OR SARS-CoV-2) AND (vaccination OR immunization) AND (paediatrics OR infants OR children OR adolescents) AND (Saudi Arabia OR KSA).

Studies were selected in accordance with these pre-defined standards (eligibility criteria):

1. Original studies which involved Saudi and non-Saudi parents, residents of KSA and demonstrating main outcomes or objectives of this review.

2. Adult participants (>18 years) who have children below 18 years in Saudi Arabia were the targeted population.

3. Studies which had the required specific survey data for pooling and analysis

4. Free full-text relevant research published in English language in peer-reviewed journals.

5. Cross-sectional observational studies.

6. Studies in the general population without subgroup-specific samples of either participating parents or their children

7. Studies published between January, 1, 2020 and October, 30, 2023.

Exclusion criteria:

1. Studies not involving Saudi parents outside KSA and not demonstrating main outcomes or objectives of this review.

2. Participants below 18 years or who do not have children below 18 years in Saudi Arabia

3. Studies not providing the required specific survey data for pooling

4. Studies which are not in free full-text, concentrating on questions irrelevant to the objectives of this SRMA or published in language other than English or published in non-peer-reviewed journals.

5. Studies other than cross-sectional studies like secondary studies without original specific research study data, SRMA, editorials, reports,, etc. . .)

6. Studies published outside the specified time range between January, 1, 2020 and October, 30, 2023.

7. Studies with specific subgroup samples of either participating parents or their children as health care workers or children with specific comorbidity.

8. Duplicate studies

S1 Table provides the excluded articles and reasons for exclusion.

## Data extraction

Two reviewers independently screened all studies for inclusion, consulting a third reviewer to resolve disagreements. After removing duplicates, the 2 reviewers independently assessed titles and abstracts. Subsequently, they applied the eligibility criteria to evaluate the full texts for potentially relevant articles. Data extraction was conducted using a standardized data extraction form (Microsoft Excel 2021), capturing relevant information from included studies, including:

- Study characteristics (authors' names, publication date, study type, study location (National or regional), research method, time/period of data collection, sample size, sampling method, and recruitment method (clinical setting or online)

- Sociodemographic characters of participants (age and gender of child, age and gender of parents, marital status, education level and occupation of parents)

- Immunization history of children and parents.

- Parents' willingness rate (%) to administer COVID-19 vaccines for their children, related reasons and associated determinants for willingness.

  S2 Table provides all data extracted from included 25 studies

  Extracted data were revised to ensure accuracy and consistency and verified by another reviewer. Stata 18 software was used for the meta-analysis.

## Risk of bias (quality) assessment

The following steps were applied to avoid or decrease risk of bias to the least level in the included studies to ensure high quality of this SRMA; selection of studies based on definite eligibility criteria and considering the reasons for excluding studies, specifying the type of included studies (e.g., cross-sectional surveys), defining the characteristics of the populations being investigated, defining and developing a comprehensive search strategy to identify relevant studies and using a PRISMA flowchart to illustrate the sequence of different phases of selection of studies for this SRMA (Fig 1).

Also, the Newcastle-Ottawa Scale (NOS) was used to assess the quality of the selected cross-sectional studies. This tool evaluates studies based on study participant selection, comparability (confounder adjustment), and outcome indicator determination with a score from 0 to 10. A "high quality" of the study was considered if the NOS score was 7 to 10 [22].

Additionally, heterogeneity was assessed between studies and evaluation of publication bias by using funnel plots to visualize potential asymmetry.

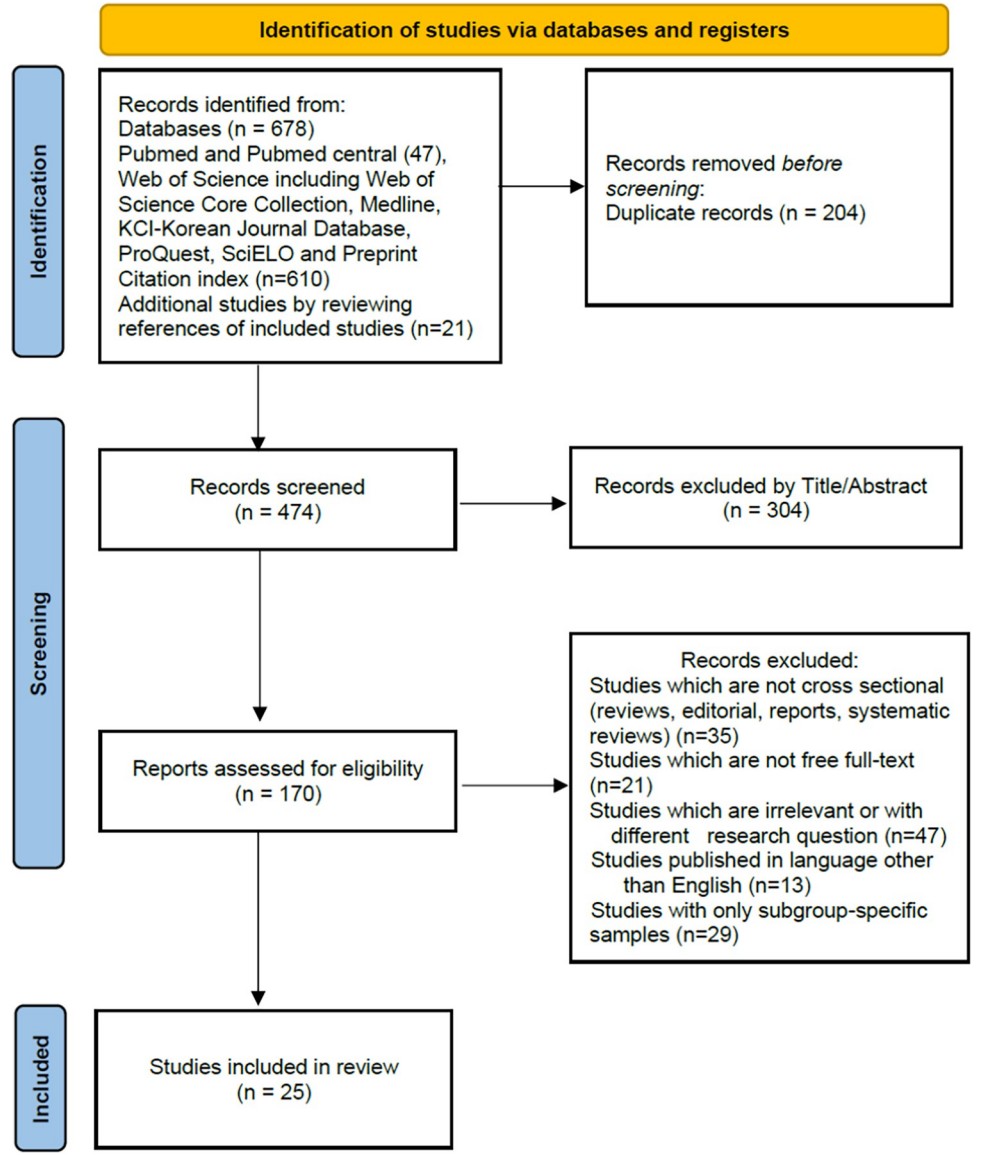

**Fig 1. Flowchart of different phases for inclusion of studies in this SRMA.**

## Statistical analysis

The meta-analysis was performed using a random-effects model to account for the expected variability across studies. The Restricted Maximum Likelihood (REML) method was applied to estimate the between-study variance, providing an unbiased estimate under the random-effects model. Forest Plot was utilized to visually summarize the individual study estimates of COVID-19 vaccine acceptance rates, along with their 95% confidence intervals and corresponding weights in the meta-analysis.

Subgroup analyses were carried out to explore potential sources of heterogeneity considering study characteristics such as sample size, region, gender of respondents, and education level. Subgroup analysis by sample size was categorized based on whether it is a small sample size with less than 1000 participants or large sample size with more than 1000 participants. Subgroup analysis by region was categorized based on whether the study covers the whole

Saudi Arabia or specific region of Saudi Arabia. Subgroup analysis by gender and education levels were categorized based on the predominant categories of the included studies (female predominance for gender and high education (university level and above) for education level). Sensitivity Analysis Leave-One-Out approach which sequentially removes one study at a time to assess the impact of any individual study on the overall effect size was used to identify studies that might be exerting a disproportionate influence on the meta-analytic summary estimate.

Galbraith plot was utilised as a graphical tool to detect potential outliers and sources of heterogeneity among studies in this meta-analysis, in this case, focusing on COVID-19 vaccine acceptance rates, by plotting standardized effect sizes against precision.

Funnel Plot was constructed to assess the risk of publication bias by plotting study effect sizes against their standard errors. A regression-based Egger test was conducted to statistically evaluate the presence of small-study effects, which could indicate publication bias.

Tests for Heterogeneity: The $I^2$ statistic was employed to quantify heterogeneity, with values greater than 75% indicating considerable heterogeneity. The Q-test was applied to assess whether the observed variance among studies was greater than expected by chance.

The Cochran's Q-test for differences between subgroups (Qb) was performed to determine if the effect sizes varied significantly across the predefined subgroups.

A qualitative analysis was used to identify and categorize the reasons influencing parental willingness to vaccinate their children against COVID-19 under recurring themes across studies to provide a deeper understanding of the contributing reasons.

For the secondary outcome of determinants/predictors or factors which affected parents' decisions to vaccinate their children with COVID-19 vaccines, Chi-squared test with calculation of odds ratios (ORs) and 95% confidence intervals (CIs) was used to compare all possibly associated factors between parents who intended and parents who clearly refused to vaccinate their children against COVID-19.

There were no missed data in this SRMA which is based on published studies and all information are included in the main text and supplementary information.

## Results

### Identification, selection and criteria of studies

After initial search, 678 records were found. Twenty-five cross-sectional studies with a total of 30,884 participants were identified according to eligibility criteria. Fig 1, presents flowchart diagram of the sequence of different phases of selection of studies for this SRMA.

Fifteen studies were nationwide from the whole kingdom of Saudi Arabia (KSA), 10 studies were region-specific mainly from the central region (7 studies), one study from the north region and 2 studies from the south region. Regarding the year of publication of studies, 6 studies were published in 2021, 14 studies published in 2022 and 5 studies were published in 2023. The convenience sampling method was applied in the majority of studies (17), snowball in 7 studies and random sampling in 3 studies. Survey method for data collection was executed by online surveys in 22 studies, and directly from parents in clinical settings like primary healthcare centres and outpatient clinics in 3 studies. All included 25 studies were of high-quality, scoring $\geq 7$ by NOS, were published in journals that apply peer-review process, and included the desired main objective of parents' willingness rate to vaccinate their children against COVID-19. Characteristics and details of the studies included in this SRMA are presented in Table 1.

**Table 1. Characteristics of the studies selected for this systematic review and meta-analysis.**

| Reference/Year of publication | Study area Location | Recruitment/Survey method | Sampling method | Sample Size | Parental willingness rate to vaccinate their children |
|---|---|---|---|---|---|
| Almusbah et al. 2021 [23] | Whole KSA Nationwide | Online | Convenience | 1000 | 256/1000 (25.6%) |
| Altulaihi et al. 2021 [24] | Riyadh Central region | Parents visiting National Guard primary healthcare centres received a self-administered questionnaire. | Convenience | 333 | 179/333 (53.7%) |
| Temsah et al. 2021 [25] | Whole KSA Nationwide | Online | Snowball | 3167 | 1507/3167 (47.6%) |
| Aldakhil et al. 2021 [26] | Riyadh Central region | Female waiting area in outpatient clinics | Non-probability purposive | 270 | 127/270 (47.0%) |
| Samannodi et al. 2021 [27] | Whole KSA Nationwide | Online | Convenience | 581 | 371/581 (63.9%) |
| Altulahi et al. 2021 [28] | Whole KSA Nationwide | Online | Convenience | 3384 | 1749/3384 (51.67%) |
| Ennaceur and Al-Mohaithef. 2022 [29] | Whole KSA Nationwide | Online | Convenience and snowball | 379 | 167/379 (44%) |
| Al-khlaiwi et al. 2022 [30] | Whole KSA Nationwide | Online | Simple random sample | 1304 | 602/1304 (46.1%) |
| Almalki et al. 2022 [31] | Whole KSA Nationwide | Online | Random sample | 4135 | 1577/4135 (38.1%) |
| Alhazza et al. 2022 [32] | Whole KSA Nationwide | Online | Convenience | 1052 | 663/1052 (63%) |
| Aljamaan et al. 2022 [33] | Whole KSA Nationwide | Online questionnaire distributed through SurveyMonkey | Convenience | 1340 | 885/1706 (51.8%) |
| Aljamaan et al. 2022 [34] | Whole KSA Nationwide | Online | Snowball | 4071 | 1711/4071 (42%) |
| Shati et al. 2022 [35] | Aseer South region | Online | Convenience | 1463 | 811/1463 (55.4%) |
| Aedh. 2022 [36] | Najran South region | Online | Convenience and snowball | 464 | 129/464 (27.8%) |
| Alenezi et al. 2022 [37] | Whole KSA Nationwide | Online | Convenience | 1340 | 442/1340 (33%) |
| Alghamdi. 2022 [38] | Whole KSA Nationwide | Online | Convenience | 123 | 28/123 (28 22.8%) |
| Almansour et al. 2022 [39] | Whole KSA Nationwide | Online | Convenience and snowball sampling | 500 | 289/500 (57.8%) |
| Al-Rasheedi et al. 2022 [40] | Qassim Central region | Online | Convenience | 597 | 198/597 (33.2%) |
| Al-Qahtani et al. 2022 [41] | Riyadh Central region | Online | Convenience | 528 | 86/528 (16.28%) |
| Khan et al. 2022 [42] | Al-Jouf North region | Online | Exponential, non-discriminatory snowball | 444 | 188/444 (42.3%) |
| Khatrawi and Sayed. 2023 [43] | Whole KSA Nationwide | Online | Snowball | 344 | 130/344 (37.8%) |
| Alalmaei Asiri et al. 2023 [21] | Whole KSA Nationwide | Online | Random sample from original list of mobile phones | 620 | 465/620 (75%) |
| Almuqbil et al. 2023 [44] | Riyadh Central region | Online | Convenience | 699 | 291/699 (41.6%) |
| Alhuzaimi et al. 2023 [45] | Riyadh Central region | Online (SurveyMonkey platform) using a QR code given to caregivers visiting vaccination centre | Convenience | 873 | 694/873 (79.5%) |
| Iqbal et al. 2023 [46] | Riyadh Central region | Both online survey and in-person distributed hard copies (whenever applicable) | Convenience | 1507 | 1352/1507 (89.7%) |

## Proportion of parents who intended to immunize their children with COVID-19 vaccines

The overall pooled proportion of parents who intended to immunize their children with COVID-19 vaccines was 48.0% (95% CI: 41.0–54.0%), across all studies (Fig 2 Forest plot of proportion of parents who intended to vaccinate their children with COVID-19 vaccines by study and year). Nearly half of the population sampled was willing to accept the vaccine. The high overall heterogeneity ($I^2$ = 99.42%) underscored the variability in acceptance rates across different studies and years. The test of group differences $Q_b(2)$ = 5.12 with p = 0.08, indicated that there was no statistically significant difference in vaccine acceptance rates across the years.

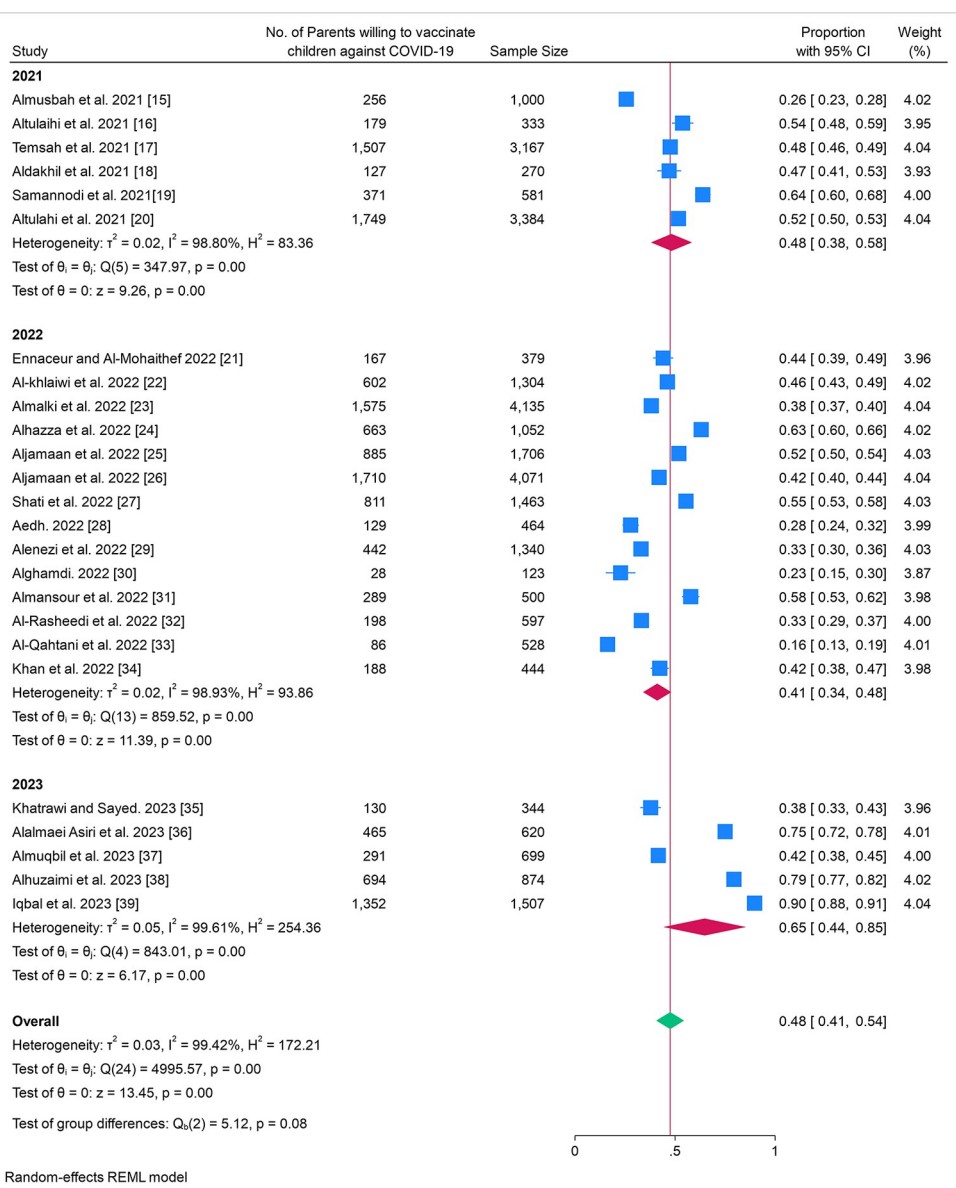

**Fig 2. Forest plot of proportion of parents who intended to vaccinate their children with COVID-19 vaccines by study and year.**

## Time trends for parental intention to immunize their children with COVID-19 vaccines

Initial meta-analysis showed that parental intention to immunize their children with COVID-19 vaccines, was correlated with study publication date with significantly increased acceptance. The overall proportion of parental acceptance for studies published in 2021 was approximately 48% (95% CI: 0.38–0.58), which was decreased to 41% (95% CI: 0.34–0.48), in studies published in 2022 and significantly increased to 65% (95% CI: 0.44–0.85), in studies published in 2023. The heterogeneity among all studies was high at 98.80%, indicating significant differences between studies that cannot be attributed to chance.

The major contribution in significantly increased acceptance rates in 2023, was related to 3 out of the 5 studies published in 2023 [36,38,39] which had the highest recorded rates among all included studies for parental intention to immunize their children with COVID-19 vaccines with recorded rates of 75%, 79.5%, and 89.7% respectively.

## Leave-One-Out Sensitivity analysis of COVID-19 vaccine acceptance rates

The sensitivity analysis presented in Fig 3, revealed a remarkable consistency in the recalculated pooled proportions of parents who accepted to immunize their children with COVID-19 vaccines, with any omitted study resulting in very similar estimates ranging narrowly between 0.46 and 0.48. These proportions were accompanied by 95% CIs that overlap considerably, reflecting a high degree of stability in the meta-analysis results. The p-values associated with each recalculated estimate were uniformly reported as 0.000, reinforcing the statistical significance of the findings despite the exclusion of individual studies. This sensitivity analysis indicated that no individual study included in the meta-analysis disproportionately skewed the overall vaccine acceptance rate. This finding enhances the credibility of the meta-analytic results, indicating that the summary estimate of COVID-19 vaccine acceptance was robust and not unduly influenced by any individual study with consistency of the observed patterns of vaccine acceptance that were reflective of a true effect.

## Galbraith plot for sources of heterogeneity and funnel plot for risk of publication bias

Fig 4: The Galbraith plot showed that most studies cluster around the regression line, suggesting a consistent effect size across these studies. However, there were a few studies that appeared to fall outside the 95% CI bands, indicating that these particular studies may be contributing disproportionately to the heterogeneity observed in the meta-analytic results.

The funnel plot presented in Fig 5 revealed a distribution of studies that appeared relatively symmetrical around the aggregate effect estimate. This symmetry indicated the absence of substantial publication bias. Regression-based Egger test was applied for quantitative assessment of publication bias within the meta-analysis of COVID-19 vaccine acceptance rates. The obtained results showed that the estimated intercept (beta1) is -6.32 with a standard error (SE) of 4.573. The corresponding z-value is -1.38, with a p-value of 0.1438. The p-value of 0.1668 indicated that there was no statistically significant evidence of small-study effects, as the p-value is above the conventional threshold of 0.05, indicating that publication bias, was not statistically significant in this meta-analysis.

## Subgroup analysis of COVID-19 vaccine parental acceptance rates by year of publication, study size, region, gender, and education level

The subgroup analysis of COVID-19 vaccine parental acceptance rates (Fig 6) revealed that there was no significant variation in vaccine acceptance rates based on year of study,

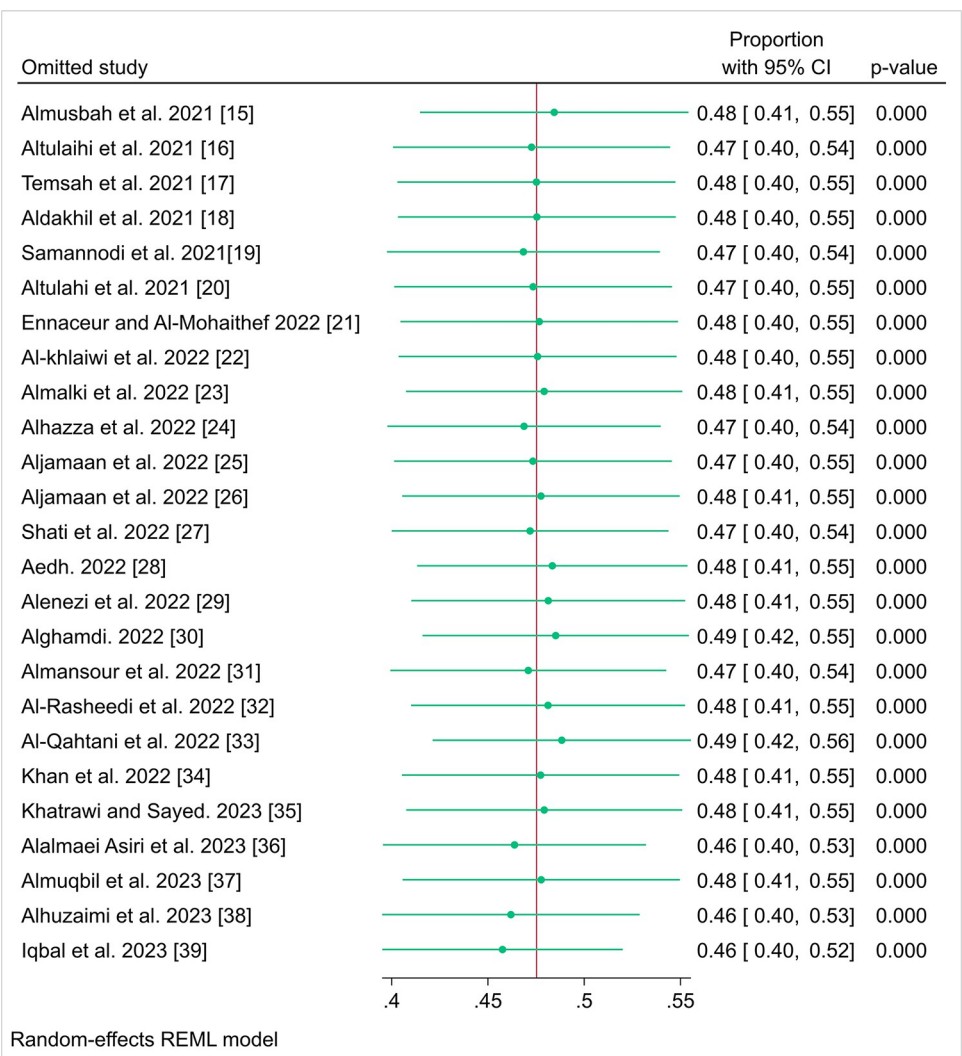

**Fig 3. Leave-one-out sensitivity analysis of COVID-19 vaccine acceptance rates: Assessing the impact of individual studies on the meta-analytic estimate.**

sample size, region, predominating gender of respondents, and education level with test for group differences ($Q_b(1)$ p values of p = 0.08, p = 0.62, p = 0.81, p = 0.51, and p = 0.50 respectively.

## Reasons of parents to immunize their children with COVID-19 vaccines

From all studies, parents declared 12 reasons to immunize their children with COVID-19 vaccines. The top 3 reasons frequently reported for parental willingness to vaccinate their children against COVID-19, were prevention of COVID-19 and protection of child, family and community from COVID-19 (11 studies), adequate information about safety and efficacy of COVID-19 vaccines (10 studies) and mandatory or compulsory vaccination ordered by government (9 studies). Table 2. provides the reasons for parental willingness to vaccinate their children against COVID-19 in descending order of frequency.

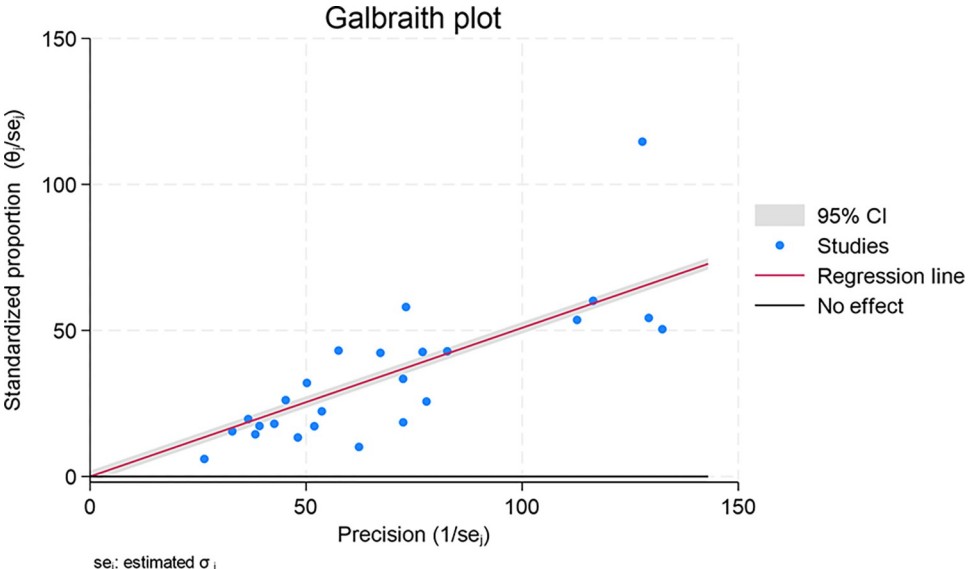

**Fig 4. Galbraith plot for assessing heterogeneity in COVID-19 vaccine acceptance meta-analysis.**

## Determinants of parents' willingness to immunize their children with COVID-19 vaccines

The determinants (factors/predictors) associated with willingness of parents to vaccinate their children with COVID-19 vaccines, were extracted by comparing all possibly associated factors between parents who intended and parents who clearly refused to vaccinate their children against COVID-19. Fifteen determinants were identified and the first 4 most significant determinants according to the estimated odds ratios and p values were; perceived efficacy of vaccines (OR: 13.44 (11.93–15.15), $p < 0.0001$), perceived safety of vaccines (3.59 (3.26–3.96, $p < 0.0001$), parents received COVID-19 vaccines (OR: 3.39 (2.69–4.28), $p < 0.0001$), and child or parent previously received influenza vaccine (3.09 (2.82–3.39), $p < 0.0001$).

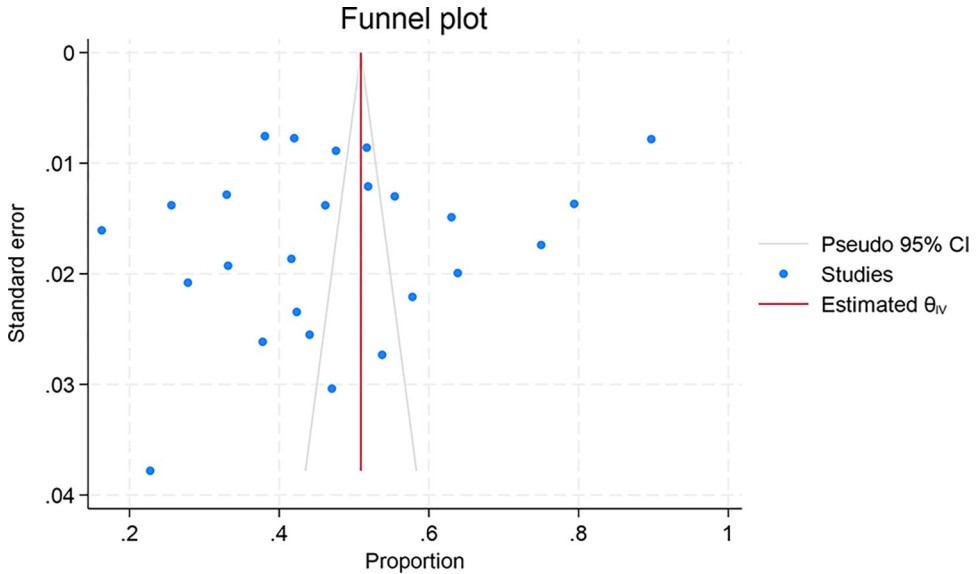

**Fig 5. Funnel plot for evaluating publication bias in the meta-analysis of COVID-19 vaccine acceptance rates.**

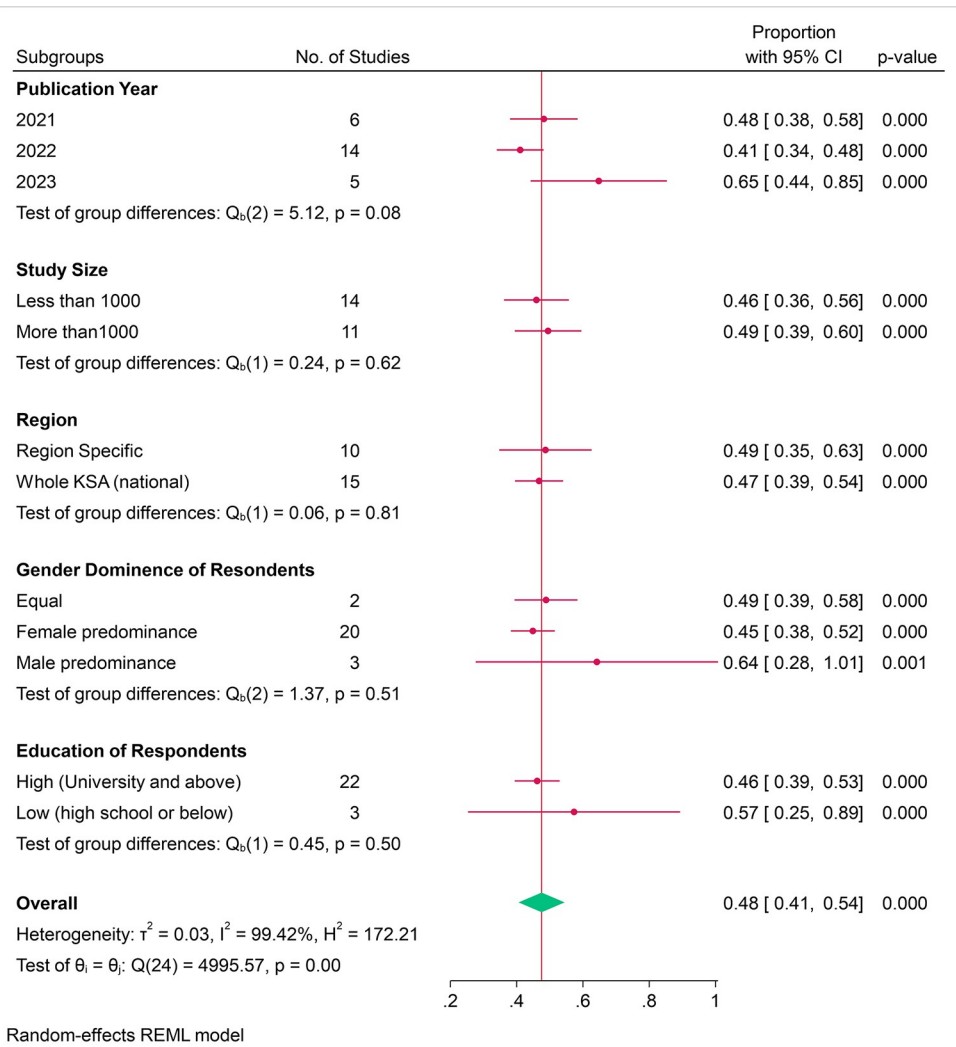

**Fig 6. Subgroup analysis of COVID-19 vaccine parental acceptance rates by year of publication, study size, region, gender, and education level.**

Other significant predictors in descending order were; fathers accepted COVID-19 vaccines more than mothers (1.74 (1.61–1.89), p< 0.0001), non-married or single parents (1.53 (1.36–1.72), p< 0.0001), parents or family member working as healthcare professional or provider (HCP) (1.34 (1.18–1.53), p< 0.0001), high-income parents (1.32 (1.20–1.44), p< 0.0001), older parents (1.29: (1.19–1.40), p< 0.0001), parents without family member infected with COVID-19 (1.16 (1.06–1.26), p = 0.0007), non-Saudi parents (1.24 (1.09–1.40), p = 0.0009) and child received previous childhood vaccines (1.64 (1.18–2.29), p = 0.003). The level of parental education and age of children were controversial or did not affect parental decision to immunize their children with COVID-19 vaccines. Table 3. shows determinants (factors/predictors) of parental willingness to immunize their children with COVID-19 vaccines. (S3 Table provides the detailed comparisons of all possible determinants (factors/predictors) of parental willingness to immunize their children with COVID-19 vaccines between parents who intended and parents who clearly refused to immunize their children with COVID-19 vaccines).

**Table 2. Reasons for parental willingness to immunize their children with COVID-19 vaccines in descending order of frequency.**

| Reason | Frequency of mentioning this reason | Studies that mentioned this reason |
|---|---|---|
| Prevention of COVID-19 and protection of child, family and community from COVID-19 | 11 | References; [21,23,26,28,30,31,35,41,43,44,46] |
| Adequate information about safety and efficacy of COVID-19 vaccines | 10 | References; [23,27,28,30,35,36,37,42,46,47] |
| Mandatory or compulsory vaccination ordered by government | 9 | References; [23,26,27,28,30,39,41,42,46] |
| Perceived susceptibility and severity of COVID-19 | 5 | References; [29,30,31,41,45] |
| Permission to enter day care centres and return to schools | 4 | References; [38,39,41,45] |
| End pandemic and return to normal life (permit work/travel) without lock down | 4 | References; [23,27,30,46] |
| Advice from healthcare worker | 4 | References; [26,28,42,43] |
| Advice and feedback from friends and family members whenever vaccines were taken by many other children | 3 | References; [27,42,47] |
| Age of child from 4–12 years | 2 | References; [30,31] |
| Decrease COVID-19 complications | 1 | Reference; [27] |
| Trust healthcare system | 1 | Reference; [29] |
| Need for vaccine against new infectious disease like COVID-19 | 1 | References; [36] |

## Discussion

Significant global morbidity and mortality occurred as a result of COVID-19 pandemic, impacting physical and mental health of children [47,48]. Given the risk to the pediatric population and ongoing transmission, vaccinating children and adolescents is recommended as the most efficient and cost-effective necessary strategy for prevention of infection, control of viral transmission in the community and potential global eradication of COVID-19 [49–51].

As legal guardians of minors, parents are responsible and authorized for making healthcare decisions for their children, including routine as well as COVID-19 vaccinations, guided by their children's best interests. Understanding parents' opinions on vaccinating their children is crucial to address their attitudes, reasons for willingness, hesitancy, and refusal. Accordingly, this SRMA was carried out to estimate parents' willingness to vaccinate their children with COVID-19 vaccines in Saudi Arabia and to identify the determinants of parental willingness for vaccination.

Up to our knowledge, this was a pioneer SRMA to assess and analyse parents' willingness to immunize their children with COVID-19 vaccines in Saudi Arabia. It was observed that the overall pooled rate of parents' willingness to administer COVID-19 vaccines for their children was 48.0% (95% CI: 41.0–54.0%). This pooled rate of parents' willingness to administer COVID-19 vaccines for their children was lower than the Saudi adult willingness rate of 52.4% [28]. Yet, it is almost identical (46.3%) to the rate found in a national survey conducted in USA [52] and in a recent systematic review to identify the barriers and facilitators affecting parents' decisions to vaccinate children aged 5–11 against COVID-19 in Western countries [16]. Also, the parental acceptance for COVID-19 vaccines for their children in this SRMA was lower than the acceptance rate of 70.0% in Chinese parents [47] and 60.1% in other SRMA from different parts of the world [15]. However, in comparison with Arab countries, the parental willingness rate to immunize children against COVID-19 detected in this SRMA was higher than the rate recorded in an online survey study from Syria where 105 out of 283 (37.1%) parents agreed to immunize their children against COVID-19 [53]. In a study from 5 Arab countries in the Middle East (Iraq, Jordan, United Arab Emirates (UAE), Oman, and Yemen), 56% of Arab parents intended to vaccinate their children with COVID-19 vaccines [54]. Moreover, a multi-country study including a sample of 3744 parents from 8 countries (Lebanon, Palestine,

**Table 3. Determinants (factors/predictors) of parental willingness to immunize their children with COVID-19 vaccines in a descending order of significance.**

| Determinant (Factors/Predictors) | Number of studies with checked determinant | Individual studies with the examined determinant | Total willing parents N % | Total refusing parents N % | OR (95% CI) | p value* |
|---|---|---|---|---|---|---|
| **Perceived efficacy of COVID-19 vaccines** <br> Effective <br> Not effective | 4 | [18,23,36,39] | 2819/3276 86.1 <br> 457/3276 13.9 | 1246/3961 31.5 <br> 2715/3961 68.5 | 13.44 (11.93–15.15) | < 0.0001 |
| **Perceived safety of COVID-19 vaccines** <br> Safe <br> Not safe | 6 | [18,21,23,28,31,39] | 1800/3311 54.4 <br> 1511/3311 45.6 | 1060/4256 24.9 <br> 3196/4256 75.1 | 3.59 (3.26–3.96) | < 0.0001 |
| **Parents received COVID-19 Vaccines** <br> Yes <br> No | 9 | [22,23,28,32,33–37] | 3574/3666 97.5 <br> 92/3666 2.5 | 4218/4586 92.0 <br> 368/4586 8.0 | 3.39 (2.69–4.28) | < 0.0001 |
| **Child or parent received influenza vaccine** <br> Yes <br> No | 6 | [15,23,32,34,35,39] | 1823/3593 50.7 <br> 1770/3593 49.3 | 1231/4929 25 <br> 3698/4929 75 | 3.09 (2.82–3.39) | < 0.0001 |
| **Gender** <br> Fathers <br> Mothers | 13 | [15,21–23,27,28,31,33–37,39] | 2001/5189 38.6 <br> 3188/5189 61.4 | 1538/5802 26.5 <br> 4264/5802 73.5 | 1.74 (1.61–1.89) | < 0.0001 |
| **Marital status** <br> Married <br> Not married/single | 10 | [21,23,27,28,31,33,34,36,37,39] | 3492/4209 83 <br> 717/4209 17 | 4460/5059 88.2 <br> 599/5059 11.8 | 1.53 (1.36–1.72) | < 0.0001 |
| **Participants or family members working as HCP** <br> Yes <br> No | 9 | [21,27,28,33–37,39] | 731/2652 27.6 <br> 1921/2652 72.4 | 504/2281 22.1 <br> 1777/2281 77.9 | 1.34 (1.18–1.53) | < 0.0001 |
| **Income** <br> Low less than 15,000 SAR <br> High more than 15,000 SAR | 9 | [21,22,23,27,28,33,36,37,39] | 2941/4288 68.6 <br> 1347/4288 31.4 | 3434/4630 74.2 <br> 1196/4630 25.8 | 1.32 (1.20–1.44) | < 0.0001 |
| **Age** <br> Older parents above 40y <br> Young parents below 40y | 10 | [21,22,23,27,28,33,35–37,39] | 1863/4654 40.0 <br> 2791/4654 60.0 | 1688/4955 34.1 <br> 3267/4955 65.9 | 1.29 (1.19–1.40) | < 0.0001 |
| **Perceived severity: infected child or family member with COVID-19** <br> Yes <br> No | 10 | [15,21–23,28,31–33,36,37] | 1534/4060 37.8 <br> 2526/4060 62.2 | 2042/4943 41.3 <br> 2901/4943 58.7 | 1.16 (1.06–1.26) | 0.0007 |
| **Nationality** <br> Saudi <br> Non-Saudi | 8 | [21,23,27,28,31,33,35,37] | 2596/3149 82.4 <br> 553/3149 17.6 | 3653/4283 85.3 <br> 630/4283 14.7 | 1.24 (1.09–1.40) | 0.0009 |
| **Child received previous childhood vaccines** <br> Yes <br> No | 3 | [15,28,33] | 391/451 86.7 <br> 60/451 13.3 | 523/655 79.8 <br> 132/655 20.2 | 1.64 (1.18–2.29) | 0.003 |

(*Continued*)

**Table 3.** (Continued)

| Determinant (Factors/Predictors) | Number of studies with checked determinant | Individual studies with the examined determinant | Total willing parents N % | Total refusing parents N % | OR (95% CI) | p value* |
|---|---|---|---|---|---|---|
| **Source of information about vaccines** Ministry of Health (MOH)/HCP Others (social media) | 4 | [18,23,31,35] | 892/2052 43.5 1160/2052 56.5 | 1201/3007 39.9 1806 60.1 | 1.16 (1.03–1.30) | 0.01 |
| **Education level** High (University and above) Low (High school and below) | 13 | [21,22,23,27,28,31–37,39] | 3038/5165 58.8 2127/5165 41.2 | 3432/5939 57.8 2507/5939 42.2 | 1.04 (0.97–1.13) | p = 0.27 |
| **Children age** Young below 5y 5-12y | 3 | [15,28,33] | 208/471 44.2 263/471 55.8 | 264/635 41.6 371/635 58.4 | 1.11 (0.87–1.41) | p = 0.39 |

* Chi-squared test.

Jordan, Iraq, Kuwait, Qatar, Saudi Arabia and the UAE), concluded that most of the parents in the Arab countries are hesitant to vaccinate their children against COVID-19 and this hesitancy can be explained by the insufficient information about vaccine safety and concerns about vaccine effectiveness [55].

The lower overall intention of parents to vaccinate their children against the COVID-19 found in this SRMA can be possibly explained by the perception of a lower risk of infection and less serious COVID-19 in children complicated by the tendency of children to be frequently asymptomatic carriers. Parents may perceive that COVID-19 vaccines are unnecessary due to relatively lower risk of COVID-19 infection and its complications in children, concerns about risks or side effects of vaccines in children, and most of other older people who need to be protected are vaccinated without need to vaccinate children. Even if parents were themselves vaccinated, many hesitated to vaccinate their children [56]. The first COVID-19 vaccine was created only recently; it's still relatively uncharted territory. Unlike parents of the 1960s, today's parents are often overwhelmed by media and internet messages, promoting misinformation about or mistrust in vaccines. It seems that occasionally the decision of parents to vaccinate self can be different when it comes to their children [57].

The great variability in parents' willingness throughout the different included studies of this SRMA was also detected in other population-specific and worldwide SRMA and could be due to different characteristics of study participants, designs of studies, attitudes towards vaccination and parental levels of information and knowledge [15,47,56,58–60]. Additionally, in Saudi Arabia, religious and cultural beliefs often intersect with healthcare decisions, including the opt to vaccination. Saudi Arabia's policy and efforts in promoting healthcare, vaccination campaigns, and endorsement of vaccinations, including COVID-19 vaccines are crucial in shaping parental attitudes and significantly influence parents' willingness to immunize their children with COVID-19 vaccines [20].

The time trends for parental willingness to administer COVID-19 vaccines for their children in relation to the time of the published studies were analysed to detect if the more recent studies express higher rate of parental willingness due to expected more available information about the efficacy and safety COVID-19 vaccines for children and consequently more trust of parents in novel COVID-19 vaccines. The analysis revealed that 3 out of the 5 studies published in 2023 [21,45,46] had the highest recorded rates among all included studies with

recorded rates of 75%, 79.5%, and 89.7% respectively. However, after more in depth analysis examining the effect of the time of data collection for these 3 studies, it was found that data collection time was not available for one study [21], one study was published after 18 months [45] and the other study was published after 13 months from data collection [46], thus these high rates of COVID-19 vaccines acceptance did not reflect actual increment in parental willingness in more recent studies published in 2023. The highest acceptance rate of 89.7% in Iqbal et al, [46], was recorded from a study that utilized both online survey and in-person distributed hard copies (whenever applicable) as a recruitment/survey method and parents were predominantly of low education level. Additionally, the analysis of time trends for parental willingness to immunize their children with COVID-19 vaccines according to data collection period could not be performed due to unavailability of data collection period for some studies and overlap of this period between many studies. So, it was concluded that the overall pooled rate of parents' willingness to administer COVID-19 vaccines for their children, is the most important to consider and interpret. Another significant finding was the remarkable heterogeneity among studies in parental willingness as evidenced by very high $I^2$ of 99.42%, which is a similarly documented finding in other SRMA [15,47,59].

Regarding the determinants/predictors of parental acceptance of COVID-19 vaccines for their children, the first 4 most significant determinants in a descending order were COVID-19 vaccines-related factors including perceived efficacy of vaccines, perceived safety of vaccines, parents received COVID-19 vaccines and child/parents received influenza vaccine.

In relation to perceived efficacy and safety of vaccines, adequate information about efficacy and safety of COVID-19 vaccines was a major reason declared by parents to accept vaccination of their children that was detected in 10 studies in this SRMA (Table 2). Similarly, a SRMA in the Italian population found that vaccine hesitancy was primarily due to a lack of information, beliefs that the vaccine was unsafe or ineffective, fear of adverse events, and the perception that COVID-19 is nonthreatening for children and adolescents [60]. Consistent with these findings, a Saudi study investigated parents' views on immunizing their children with COVID-19 vaccines. The study found that 98.2% of parents indicated that they wanted information about the safety of vaccine in children to make an informed decision [61]. Similarly, the fear of side effects was the most important reason of not vaccinating children in a multinational scoping review [62].

Parents who received COVID-19 vaccines appear to have the attitude of more acceptance towards vaccinating their children. This is likely due to reduced concerns after experiencing no serious side effects themselves. Similar behaviours have been noted in other studies [15,60]. Moreover, a history of vaccination against influenza in children or parents positively influenced parents' intention to give their children COID-19 vaccines [15]. In this SRMA, it was observed that previous influenza vaccination was more significantly associated with parental willingness to have their children receive the COVID-19 vaccine compared to routine childhood vaccinations. This may be due to parents having greater trust, confidence and acceptance of well-known vaccines that prevented infections, deriving them to opt for non-mandatory or vaccine that may be regarded as unnecessary like influenza vaccine and, similarly, the new COVID-19 vaccines. One recent Chinese study found that parents' hesitancy to receive influenza vaccine for themselves was associated with a higher level of parental hesitancy to vaccinate their children against influenza [63].

Other significant predictors/determinants included fathers, non-married or single parents, participants or family members working as HCP, parents with high income, older parents above 40 years, parents without family member infected with COVID-19, non-Saudi parents, child received previous childhood vaccines and parents who have information about COVID-19 vaccines from MOH/HCP)

Consistent with these results, the finding that fathers accepted COVID-19 vaccination of their children more than mothers has been demonstrated in many studies. In 7 out of 24 studies of a SRMA, fathers were more likely willing than mothers to vaccinate their children, while 17 studies found no significant association [15]. Also, in a study from Taiwan, 67.5% of mothers compared to 50% of fathers were hesitant to immunize their children with COVID-19 vaccines [64]. This could be attributed to that mothers might be more concerned about potential adverse effects of the COVID-19 vaccines for themselves with possible fears from effects of vaccines on their general health, fertility, pregnancy, and breastfeeding as well as adverse effects in their children. Furthermore, mothers may have strong feelings and mainly responsible for care of their children, so, mothers, as directors of family health decisions, often avoid risks they perceive as uncontrollable, viewing vaccines as such a risk [65].

In this SRMA, participating parents or family members working as HCP positively influenced parental willingness to immunize their children with COVID-19 vaccines. Parents working as HCP have better access to information about vaccines, which helps them avoid conspiracy theories and makes them more likely to vaccinate their children. However, they often express concerns about the safety, efficacy, and unknown side effects of newly developed vaccines [47]. The confidence in HCP as reliable sources for information about COVID-19 vaccine strongly predicted vaccine acceptance. On the other hand, hesitancy of HCP may reduce parents' intention to vaccinate their children. The child's doctor or HCP was considered as the most trusted source of information about COVID-19 vaccines [47,60,66,67].

Similar to our findings, in a SRMA, higher income of parents was linked to their intention to vaccinate their children in 7 out of 20 studies, while 2 studies found the opposite relationship and 11 studies found no effect of income [15]. According to this SRMA, older parents were more willing to immunize their children with COVID-19 vaccines. Similar findings were reported in 10/25 studies of a worldwide SRMA and in a Chinese SRMA [15,67].

In this SRMA, parents without a family member infected with COVID-19 had significantly more desire to immunize their children with COVID-19 vaccines compared to parents who had a family member infected with COVID-19. This may be explained by more careful parents to prevent COVID-19 in their children or more fears from a possible worse scenario of COVID-19- associated infection with fatal outcome while parents who had a family member infected with COVID-19 most likely experienced mild or moderately severe COVID-19 infection without complications. However, among Italian parents, experiencing COVID-19 or previous infections among family members and friends, appeared to increase the intention to vaccinate children [60].

In the current SRMA, non-Saudi parents were significantly more willing to immunize their children with COVID-19 vaccines than Saudi parents. This may be related to more fears if their children acquired COVID-19 infection and its complications with concerns about receiving adequate treatment away from their home country. Moreover, non-Saudi parents may have experienced high pressure and psychological stress encountered by expatriate families to receive mandatory COVID-19 vaccines to maintain their work and activities, enter public places and have vaccine passport necessary for travelling.

Moreover, in this SRMA, parents who had their information about COVID-19 vaccines from governmental source as MOH and HCP were significantly more willing to immunize their children with COVID-19 vaccines compared to parents who got their information from social media and unofficial sources like personal beliefs, family relatives/friends' opinions, Internet, and television. This is considered as an advantage showing trust of parents in Saudia Arabia in the provided healthcare services and information about COVID-19 vaccines distributed by official sources like MOH and HCP. It is worthy to mention that the role of social media during COVID-19 pandemic is very critical because people were occasionally or

consistently exposed to variable and conflicting information about COVID-19 vaccination across different social media platforms. Thus, the effect of social media has been explored in many studies with an important conclusion that individuals who relied on the Internet or social media as their primary source of information exhibited higher levels of COVID-19 vaccine hesitancy, due to the prevalence of misleading, inaccurate, and unverified information disseminating online [47,60,68,69].

In this SRMA, education level as one of the sociodemographic predictors, could not influence parental decision towards immunizing their children which is similar to the result from another SRMA in China [67]. The level of education has always been a controversial issue. Some studies found that higher education and access to scientific information were linked to greater vaccine acceptance [60]. In a SRMA, 15 out of 29 studies showed that higher education was associated with increased vaccination intent, 6 studied showed that lower education had this association, and 8 studies found no link [15]. Another SRMA revealed that Chinese parents with lower education levels were more likely to immunize their children with COVID-19 vaccines than those with higher education [47]. This inconsistent finding may be related to different characteristics of studies with large variability in sample size and percentage of parents with high versus low educational level.

Actually, in this SRMA, vaccine-related factors including perceived efficacy of vaccines, perceived safety of vaccines, parents received COVID-19 vaccines, and child or parents received influenza vaccine were more significant determinants in this respect. Therefore, this SRMA emphasizes the Priority to Focus on vaccine-related factors as main/key strategy or target of COVID-19 vaccines' drivers to convince parents in a logical way based on sound accurate transparent cumulative and emerging scientific data about efficacy and safety of COVID-19 vaccines to optimize their uptake for children and adolescents. The lower risk of serious illness in healthy children means that even minor vaccination risks must be considered. This SRMA suggested that well-informed parents tend to be less fearful and anxious about COVID-19 vaccines and are more likely to vaccinate their children.

Fortunately, the role of MOH and HCP as trustable sources for COVID-19 vaccines information was significant and appreciated in this SRMA which provides a good opportunity of such official sources to communicate with parents effectively to deliver clear messages and recommendations about COVID-19 vaccines for children.

This SRMA had some limitations. At first glance, the main limitation of this SRMA was the high heterogeneity among the included studies. However, in fact, this remarkable heterogeneity may not be an actual limitation as it was consistently and universally reported in similar SRMA [15,47,60] which can reflect that the decision of parents to administer COVID-19 vaccines for their children is a very complex process influenced by many factors/determinants with marked variability among different studies in different populations as well as within the same population regardless of the duration of the time that has elapsed since the evolution of novel COVID-19 vaccines and cumulative data of their proven efficacy and safety in children and adolescents. Moreover, a random-effects model analysis, subgroup analysis and leave-one-out sensitivity analysis were used to minimize this bias and increase the robustness of results. Second, most studies used convenience sampling, potentially affecting representativeness of population due to the lack of random sampling. Additionally, many studies relied on online surveys or questionnaires, which may exclude those without Internet access to the provided questionnaire. Third, the overall pooled rate of parents' willingness to immunize their children with COVID-19 vaccines in Saudi Arabia (48%) is subjective and reflects a general trend but can fluctuate based on various factors including changes in vaccination policies, the evolving public perception of the vaccine's safety and efficacy, and the perceived risk of infection. Unlike fixed vaccination uptake rates, willingness can be more dynamic and influenced by

ongoing informational changes. Fourth, the analysis of parental acceptance according to data collection period could not be performed due to unavailability of data collection period for some studies and overlap of this period between many studies. Finally, only studies published until October, 2023, were included. However, as new data/evidence from randomized controlled trials are increasing on ongoing basis in a dynamic process, and the availability of COVID-19 vaccines evolves, parents' attitudes towards vaccination can shift.

However, this SRMA has considerable strengths. Up to our knowledge, this is a pioneer SRMA to assess and analyse parents' willingness to immunize their children with COVID-19 vaccines in Saudi Arabia and to identify determinants of vaccine willingness. Indeed, the main findings of this SRMA with suboptimal overall parental willingness rate to immunize their children which was influenced more significantly by the vaccine-related factors, can provide valuable insights for policymakers and HCP. These insights can guide the development of evidence-based policies to improve parental willingness to vaccinate children, which is crucial for controlling SARS-CoV-2 spread and promoting herd immunity in the community particularly if the virus continues to pose a major threat not only in Saudi Arabia but the results may be generalized globally because the most important main findings of this SRMA, also emphasized similar findings reported in previous studies. Additionally, this SRMA may predict the expected attitude/behaviour of parents in similar emerging future global pandemic threats. Another strength was focusing on willingness/intention to vaccinate which was mentioned in all included studies rather than vaccine hesitancy which has variable definitions among different studies. Furthermore, the adequate number of included studies (25) which was more than the number of studies included in SRMA of vaccine willingness/hesitancy in some population-based national studies [47,60], with a sample size of 30,844 participants resulting from the selected studies allowed us to perform subgroup analysis and extract determinants of vaccine acceptance among parents of minors in Saudi Arabia. Also, the remarkable efforts performed to extract and compare all possibly associated factors between parents who intended and parents who clearly refused to vaccinate their children against COVID-19, with calculations of ORs and CIs, allowed for highly accurate quantitative measurement of the magnitude/role of each determinant associated with parental willingness to immunize their children with COVID-19 vaccines. Additionally, excluding pre-print articles and including studies published in peer-reviewed journals and high-quality studies according to NOS, helped to ensure the quality and reliability of the data of this SRMA. Finally, the risk of bias and publication bias were considered and evaluated to ensure that the results of this SRMA are robust and not unduly influenced by methodological flaws or selective reporting.

This SRMA has important implications mainly addressing safety concerns, countering misinformation, and building vaccine confidence as vital determinants of COVID-19 vaccines' acceptance for children. This involves offering clear accurate evidence-based scientific information on vaccine efficacy and safety, engaging with communities to understand their concerns, and partnering with trusted organizations to disseminate accurate information.

## Conclusions

The overall pooled rate of parents who intended to immunize their children with COVID-19 vaccines in Saudi Arabia, was 48.0% (95% CI: 41.0–54.0%) with high heterogeneity ($I^2$ = 99.42%) across different studies and years. The main reason for parents to vaccinate children was to prevent COVID-19 and protect child, family and community from COVID-19. The top 4 most significant determinants associated with parents' willingness to vaccinate children were perceived efficacy/safety of vaccines, parents received COVID-19 vaccines, and child or parents received influenza vaccine. Therefore, this was the first SRMA from Saudi Arabia

which emphasized the priority to focus on vaccine-related factors as main/key strategy of COVID-19 vaccines' drivers to convince parents in a logical way based on accurate cumulative and emerging scientific data about efficacy and safety of COVID-19 vaccines to optimize their uptake by children/adolescents. This SRMA can provide valuable insights for development of evidence-based policies to improve parents' confidence/trust in vaccines and consequently their willingness to vaccinate their children, which is crucial for controlling SARS-CoV-2 spread and promoting herd immunity in the community particularly if the virus continues to pose a major threat.

## Recommendations

Physicians and other reliable HCP have duties to help control the transmission of diseases like COVID-19 in the community and achieve the target of herd immunity by practicing 2 important roles: one as a caregiver and the other as a communicator, both of which can save lives. This is extremely important because the world may face more pandemics, and the technology to rapidly create vaccines that can preserve lives is available but a vaccine is useless if people refuse to take it. Finally, further research is needed to follow the changes in parents' attitude, behavior and willingness to vaccinate their children against COVID-19 and to capture the most significant reasons and determinants of parents' intentions.

Furthermore, social media should be a powerful tool to spread accurate, easily understood information to enhance vaccine trust and accessibility. This can boost parents' confidence and willingness to immunize their children with COVID-19 vaccines. However, extensively forceful messages and campaigns can backfire, instilling fear and leading to the dismissal or suppression of the information.

## Supporting information

**S1 Checklist. PRISMA 2020 checklist.**
(DOCX)

**S1 Table. Excluded articles and reasons for exclusion.**
(PDF)

**S2 Table. Data extracted from included 25 studies.**
(XLSX)

**S3 Table. Comparison of all possible determinants (factors/predictors) of parental willingness to immunize their children with COVID-19 vaccines between parents who intended and parents who clearly refused to immunize their children with COVID-19 vaccines.**
(PDF)

## Acknowledgments

Authors thank Miss. Aya Moustafa Hegazi for her help in English language editing.

## Author Contributions

**Conceptualization:** Moustafa Abdelaal Hegazi, Mohamed Hesham Sayed, Nadeem Shafique Butt, Turki Saad Alahmadi, Nadeem Alam Zubairi, Wesam Abdelaziz Elson.

**Data curation:** Moustafa Abdelaal Hegazi, Nadeem Shafique Butt.

**Formal analysis:** Moustafa Abdelaal Hegazi, Mohamed Hesham Sayed, Nadeem Shafique Butt, Turki Saad Alahmadi, Nadeem Alam Zubairi, Wesam Abdelaziz Elson.

**Investigation:** Moustafa Abdelaal Hegazi, Mohamed Hesham Sayed, Nadeem Shafique Butt, Turki Saad Alahmadi.

**Methodology:** Moustafa Abdelaal Hegazi, Mohamed Hesham Sayed, Nadeem Shafique Butt, Turki Saad Alahmadi, Nadeem Alam Zubairi, Wesam Abdelaziz Elson.

**Project administration:** Moustafa Abdelaal Hegazi.

**Supervision:** Moustafa Abdelaal Hegazi.

**Validation:** Moustafa Abdelaal Hegazi, Mohamed Hesham Sayed, Nadeem Shafique Butt.

**Visualization:** Moustafa Abdelaal Hegazi, Nadeem Shafique Butt, Turki Saad Alahmadi, Nadeem Alam Zubairi, Wesam Abdelaziz Elson.

**Writing – original draft:** Moustafa Abdelaal Hegazi, Mohamed Hesham Sayed, Nadeem Shafique Butt, Turki Saad Alahmadi, Nadeem Alam Zubairi, Wesam Abdelaziz Elson.

**Writing – review & editing:** Moustafa Abdelaal Hegazi, Mohamed Hesham Sayed, Nadeem Shafique Butt, Turki Saad Alahmadi, Nadeem Alam Zubairi, Wesam Abdelaziz Elson.

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
