## [Decision Letter · Decision Letter 0]

20 Dec 2024

PONE-D-24-31975Navigating the shots: Parental willingness to immunize their children with COVID-19 vaccines in Saudi Arabia explored through a systematic review and meta-analysisPLOS ONE

Dear Dr. Hegazi,

Thank you for submitting your manuscript to PLOS ONE. After careful consideration, we feel that it has merit but does not fully meet PLOS ONE’s publication criteria as it currently stands. Therefore, we invite you to submit a revised version of the manuscript that addresses the points raised during the review process.

We look forward to receiving your revised manuscript.

Kind regards,

Maher Abdelraheim Titi

Academic Editor

PLOS ONE

Journal Requirements:

3. As required by our policy on Data Availability, please ensure your manuscript or supplementary information includes the following: 

Reviewers' comments:

Reviewer's Responses to Questions

**Comments to the Author**

1. Is the manuscript technically sound, and do the data support the conclusions?

Reviewer #1: Yes

Reviewer #2: Yes

Reviewer #3: Yes

Reviewer #4: Yes

2. Has the statistical analysis been performed appropriately and rigorously? 

Reviewer #1: Yes

Reviewer #2: Yes

Reviewer #3: Yes

Reviewer #4: Yes

3. Have the authors made all data underlying the findings in their manuscript fully available?

Reviewer #1: Yes

Reviewer #2: Yes

Reviewer #3: No

Reviewer #4: Yes

4. Is the manuscript presented in an intelligible fashion and written in standard English?

Reviewer #1: Yes

Reviewer #2: Yes

Reviewer #3: No

Reviewer #4: Yes

5. Review Comments to the Author

Reviewer #1: You have submitted a manuscript with significant information regarding vaccine uptake in Saudi Arabia. In my view, the discussion is quite long. I dont have any other suggestions. Did you think it could be shortened?

Reviewer #2: Title

Excellent. I’d propose a slight modification: Navigating the shots: A systematic review exploring parental willingness to immunize their children with COVID-19 vaccines in Saudi Arabia.

Abstract

Very well written with a clear summary for each of the main sections of the report.

Introduction

In lines 120-121 the authors state: “This lower intention may stem from perceptions that COVID-19 is less severe in children and concerns about vaccine safety in this age group and complicated by the tendency of children to be frequently asymptomatic carriers [11,12].” Here, it would be beneficial to note that a recent systematic review exploring parental barriers and facilitators in Western countries identified exactly that. Davey and Gaffiero (2024) identified key barriers preventing COVID-19 vaccination including concerns about side effects. This would be useful to include to enable comparison to the global context in the discussion.

See: Davey, S. A., & Gaffiero, D. (2024). COVID-19 vaccination in children aged 5–11: a systematic review of parental barriers and facilitators in Western countries. Therapeutic Advances in Vaccines and Immunotherapy, 12, 25151355241288115.

Moreover, it would also be beneficial to briefly consider other potential factors influencing lower vaccination intention (as this has only been briefly touched upon but is vital), such as vaccine hesitancy, parental age and greater perceived barriers. See:

Salazar, T. L., Pollard, D. L., Pina-Thomas, D. M., & Benton, M. J. (2022). Parental vaccine hesitancy and concerns regarding the COVID-19 virus. Journal of Pediatric Nursing, 65, 10-15.

Davey, S. A., Hampson, C., Christodoulaki, M. E., & Gaffiero, D. (2024). Investigating the predictors of COVID-19 vaccine decision-making among parents of children aged 5–11 in the UK. Vaccine.

In lines 129-134 the authors highlight the contribution of their work – however, this is actually understated and it’s importance should be highlighted further. For example, by identifying the factors that influence parents’ willingness to administer COVID-19 vaccines for their children, this can inform targeted public health interventions to tackle any perceived barriers and utilise any facilitators, this should be clearly stated.

Methodology

Would the primary questions not be better placed at the end of the introduction section?

This section is very well-written.

That said, could authors please elaborate on the ‘analysed using appropriate software’ – please be specific here (lines 183-184).

With respect to inclusion/exclusion criteria, did the authors also include children with chronic health conditions?

In terms of sociodemographic characteristics (lines 178-179) please specify the full characteristics as opposed to using ‘ect’.

Results

The figure 1 flow chart all adds up correctly and is well explained. Overall, the results are excellently written, presented in the tables and figures very well and this was a pleasure of a section to read. It is well-structured and the analyses employed with respect to the meta-analysis are robust and well considered. Moreover, the analysis of key determinants from the studies is neat.

Discussion

A citation is needed to support the claim made in lines 433-435.

Please avoid very short paragraphs as in line 443 – 445. This can be merged into the prior paragraph.

It would also be worth incorporating parental willingness as reported by Davey and Gaffiero (2024) who observed only 46.35% of parents intended to vaccinate their children aged 5–11 against COVID-19 (lines 451 and 545).

Excellent discussion is provided in lines 465 – 476. I commend the authors for this.

Merge the paragraph in lines 477-481 to the previous paragraph. It will aid in flow and readability.

In lines 506-512 please remove the statistical information – it is not required.

Line 538 – please revise ‘In consistence’ with ‘Consistent with these results’.

Please again try to avoid very short paragraphs as in lines 565-567, 589-591 634-637.

A section on the implications of your findings would be better placed prior to the conclusion. Moreover, can the authors offer any insight as to how such factors may influence potential future vaccination decision-making in a post-pandemic landscape? What can we learn from the findings to inform future health promotion efforts? Can the authors also differentiate between factors that influence parental COVID-19 vaccination globally, but also factors specific to Saudi Arabia?

Overall, this was a pleasure to read and review. I commend the authors on their hard work in the production of this high quality manuscript.

Reviewer #3: Thank you for the opportunity to review this interesting and well-written paper. I appreciate the effort put into this study. However, I have several concerns and suggestions regarding the methodology and discussion sections:

Line 145: Consider replacing the term "reasons" with "factors" for greater precision.

Line 149: It is unclear why the authors did not include Scopus in the search strategy. Please provide a rationale for this choice.

Line 207: When conducting the subgroup analysis, how were the factors (e.g., sample size, region, gender of respondents, and education level) defined and categorized? Further elaboration would strengthen the methodology section.

Line 218: The term "I²" should be italicized for consistency with reporting standards.

Meta-analysis software: Could you specify the software used to perform the meta-analysis? This information would enhance the reproducibility of the study.

PRISMA Checklist: Have you utilized the PRISMA Checklist to guide the reporting of your systematic review? Including this would align the study with best practices.

Line 471: The finding that "Even if parents were themselves vaccinated, many hesitated to vaccinate their children" is intriguing and contrasts with existing literature (e.g., [DOI: 10.1080/21645515.2024.2333111]). Could you expand on potential explanations for this discrepancy?

Line 524: Consider incorporating additional relevant literature, such as [DOI: 10.1186/s12916-024-03538-1], to strengthen this section.

Line 518: Regarding vaccine efficacy, it might be useful to reference similar studies, such as [DOI: 10.3390/vaccines12090988].

Discussion length: The discussion section is quite lengthy. I recommend shortening it to improve focus and readability.

Reviewer #4: Thanks for the invitation to review the manuscript. This is a well-conducted and well-written SRMA that aimed to study parental willingness to immunize their children with COVID-19 vaccines in Saudi Arabia. The following are suggestions for the authors to consider to improve the manuscript.

General issue:

- The full spelling should be given when an abbreviation is introduced for the first time, e.g., COVID-19, CDC.

- Please cite the reference(s) where appropriate when the contents are about the results of previous studies.

Introduction

- Line 79-82: This paragraph could be combined with the first paragraph. Also, the data cited were about the situation in the United States. Are there any data related to the study place, i.e., Saudi Arabia?

- Line 93-95 & 108-111: Please provide reference(s).

- Please mention a few studies about parents’ intention to vaccinate their children against COVID-19 which were conducted in Saudi Arabia.

- The knowledge gap for conducting the current SRMA is unclear. Why is an SRMA important for Saudi Arabia despite the available global SRMA? Are there any cultural or regional differences that may contribute to the differences in parents’ intentions in Saudi Arabia compared with other places?

Method

- Line 141-142: Since “PICO” was mentioned, please describe what they are in the context of the current SRMA.

- The databases mentioned were inconsistent between the abstract and methods.

- Line 159-167: Only inclusion criteria were mentioned. Please also give the exclusion criteria.

Results

- The flowchart of Figure 1 should have 4 phases. Please add “Eligibility” between “Screening” and “Included”.

Discussion

- Line 656-657: Please explain briefly for the fourth limitation here.

Look forward to reading the revised version of this manuscript.

6. PLOS authors have the option to publish the peer review history of their article (what does this mean?). If published, this will include your full peer review and any attached files.

Reviewer #1: No

Reviewer #2: **Yes: **Daniel Gaffiero

Reviewer #3: **Yes: **Siyu Chen

Reviewer #4: No

---

## [Author Response · Author response to Decision Letter 0]

4 Jan 2025

Dear Dr. Maher Abdelraheim, Academic Editor and Editor in Chief for PLOS ONE:

Thank you so much for thorough and robust revision of the Submission: PONE-D-24-31975 by 4 reviewers to ensure high quality of this systematic review and meta-analysis (SRMA). Many thanks to the reviewers who appreciated the effort put into this SRMA and considered it as an interesting and well-conducted with well-written sections. Again, thank you so much for giving us the chance to respond, reply to all concerns of reviewers, and make the necessary amendments to improve the written manuscript.

.

Authors would like to tell that the following items are included in the revised submission: 

A rebuttal letter that responds to each point raised by the academic editor and reviewer(s). and this letter is upload as a separate file labelled ‘Response to Reviewers’.

A marked-up copy of the revised manuscript that highlights changes made to the original version in blue colour. The revised manuscript with highlights is uploaded as a separate file labelled ‘Revised Manuscript with Track Changes’.

An unmarked version of the revised paper without tracked changes is uploaded as a separate file labelled ‘Manuscript’.

The revision of manuscript has been completed within the requested time frame for revision.

Reply to Editor’s Comments:

Authors would like to confirm that the manuscript was revised according to PLOS ONE’s Journal Requirements.

1. The revised manuscript meets PLOS ONE’s style requirements, including those for file naming.

2. Captions were included for Supporting Information files at the end of the manuscript, and any in-text citations were updated to match accordingly. 

3. As required by PLOS ONE’s policy on Data Availability, the manuscript and supplementary information included:

-A numbered table of all studies identified in the literature search, including those that were excluded from the analyses. For every excluded study, the reason(s) for exclusion were listed in the table. (S1 Table presented excluded studies and reasons for exclusion). 

-A table (S2 Table) of all data extracted from the primary research sources for the systematic review and/or meta-analysis. The S2 table included the following information for each study: 

-Name of data extractors and date of data extraction 

-Confirmation that the study was eligible to be included in the review. 

-All data extracted from each study for the reported systematic review and/or meta-analysis that would be needed to replicate analyses. 

-The completed risk of bias and quality/certainty assessments for each study or outcome. 

Regarding, handling of missed data, this is not applicable in this SRMA which is based on published studies and all information are included in the main text and supplementary information. 

Reply to Reviewer 1 Comments:

Reviewer #1: You have submitted a manuscript with significant information regarding vaccine uptake in Saudi Arabia. In my view, the discussion is quite long. I don’t have any other suggestions. Did you think it could be shortened?

Response/Reply:

Thanks to the reviewer for this valuable comment. 

As advised, the discussion has been shortened.

Reviewer #2: Title

Excellent. I’d propose a slight modification: Navigating the shots: A systematic review exploring parental willingness to immunize their children with COVID-19 vaccines in Saudi Arabia.

Navigating the shots: Parental willingness to immunize their children with COVID-19 vaccines in Saudi Arabia explored through a systematic review and meta-analysis

Response/Reply:

Thanks a lot, to the reviewer for acknowledging the title of this SRMA.

Authors would like to emphasize that it is essential to retain the original title even if it is slightly longer by only 2 words to include a very critical word (meta-analysis) which is a major and unique section of this SRMA which was highly appreciated and acknowledged by other reviewers and differentiating this SRMA from any other similar systematic review without meta-analysis. 

Abstract

Very well written with a clear summary for each of the main sections of the report.

Thanks a lot, to the reviewer for acknowledging the abstract of this SRMA as well-written with clear summary for each of the main sections of the report. 

Introduction

In lines 120-121 the authors state: “This lower intention may stem from perceptions that COVID-19 is less severe in children and concerns about vaccine safety in this age group and complicated by the tendency of children to be frequently asymptomatic carriers [11,12].” 

Here, it would be beneficial to note that a recent systematic review exploring parental barriers and facilitators in Western countries identified exactly that. Davey and Gaffiero (2024) identified key barriers preventing COVID-19 vaccination including concerns about side effects. This would be useful to include to enable comparison to the global context in the discussion.

See: Davey, S. A., & Gaffiero, D. (2024). COVID-19 vaccination in children aged 5–11: a systematic review of parental barriers and facilitators in Western countries. Therapeutic Advances in Vaccines and Immunotherapy, 12, 25151355241288115.

Moreover, it would also be beneficial to briefly consider other potential factors influencing lower vaccination intention (as this has only been briefly touched upon but is vital), such as vaccine hesitancy, parental age and greater perceived barriers. See:

Salazar, T. L., Pollard, D. L., Pina-Thomas, D. M., & Benton, M. J. (2022). Parental vaccine hesitancy and concerns regarding the COVID-19 virus. Journal of Pediatric Nursing, 65, 10-15.

Davey, S. A., Hampson, C., Christodoulaki, M. E., & Gaffiero, D. (2024). Investigating the predictors of COVID-19 vaccine decision-making among parents of children aged 5–11 in the UK. Vaccine.

Response/Reply:

Thanks to the reviewer for these notes and valuable input. 

The recent systematic review exploring parental barriers and facilitators in Western countries by Davey and Gaffiero (2024) which identified key barriers preventing COVID-19 vaccination including concerns about side effects and other potential factors influencing lower vaccination intention, has been considered and included in the revised manuscript.

Other potential factors influencing lower vaccination intention, such as vaccine hesitancy, parental age and greater perceived barriers were considered with making use of the provided beneficial references. 

In lines 129-134 the authors highlight the contribution of their work – however, this is actually understated and it’s importance should be highlighted further. For example, by identifying the factors that influence parents’ willingness to administer COVID-19 vaccines for their children, this can inform targeted public health interventions to tackle any perceived barriers and utilise any facilitators, this should be clearly stated.

Response/Reply:

Thanks a lot for the reviewer for these notes and valuable input. 

The contribution of this SRMA is further highlighted. It has been clearly stated that identifying the factors that influence parents’ willingness to administer COVID-19 vaccines for their children, can inform targeted public health interventions to tackle any perceived barriers and utilise any facilitators.

Methodology

Would the primary questions not be better placed at the end of the introduction section?

Response/Reply:

The primary questions (aims of this SRMA) were placed at the end of the introduction section as it was stated that this SRMA aimed to estimate parents' willingness to immunize their children with COVID-19 vaccines in Saudi Arabia and to identify the reasons and determinants influencing parents' decisions. This is as usual to mention shortly the aims/objectives of a study at the end of introduction and to elaborate more specifically on primary and secondary questions/outcomes of the study in the methodology. 

This section is very well-written.

Response/Reply

Thanks a lot, to the reviewer for acknowledging the Methodology section as well-written. 

Could authors please elaborate on the ‘analysed using appropriate software’ – please be specific here (lines 183-184)

Response/Reply:

Stata 18 software was used for the Meta Analysis

With respect to inclusion/exclusion criteria, did the authors also include children with chronic health conditions?

Response/Reply:

Children with chronic health conditions were not included as it was mentioned in the PRISMA flowchart that studies with only subgroup-specific samples (either specific subgroup of participating parents or their children) were not eligible or excluded from this SRMA which was conducted to achieve the required objectives/outcomes in the general population of parents with healthy children who constitute the majority of the community. This was also added in the inclusion criteria in the revised manuscript. 

In terms of sociodemographic characteristics (lines 178-179) please specify the full characteristics as opposed to using ‘ect’.

Response/Reply:

The full sociodemographic characteristics have been characterized instead of using ‘etc’. 

Results

The figure 1 flow chart all adds up correctly and is well explained. Overall, the results are excellently written, presented in the tables and figures very well and this was a pleasure of a section to read. It is well-structured and the analyses employed with respect to the meta-analysis are robust and well considered. Moreover, the analysis of key determinants from the studies is neat.

Response/Reply:

Thanks a lot, to the reviewer for acknowledging and appreciating the results section.

Discussion

A citation is needed to support the claim made in lines 433-435.

Response/Reply:

This section in lines 433-435 is removed in the revised manuscript to shorten the discussion as recommended by other reviewers. 

Please avoid very short paragraphs as in line 443 – 445. This can be merged into the prior paragraph.

Response/Reply:

Short paragraphs as in line 443 – 445 were avoided and merged into the prior paragraph.

It would also be worth incorporating parental willingness as reported by Davey and Gaffiero (2024) who observed only 46.35% of parents intended to vaccinate their children aged 5–11 against COVID-19 (lines 451 and 545).

Response/Reply:

Parental willingness as reported by Davey and Gaffiero (2024) who observed only 46.35% of parents intended to vaccinate their children aged 5–11 against COVID-19, was incorporated.

Excellent discussion is provided in lines 465 – 476. I commend the authors for this.

Response/Reply:

Thanks a lot, to the reviewer for his commendation. 

Merge the paragraph in lines 477-481 to the previous paragraph. It will aid in flow and readability.

Response/Reply:

The paragraph in lines 477-481 was merged to the previous paragraph to aid in flow and readability.

In lines 506-512 please remove the statistical information – it is not required.

Response/Reply:

The statistical information in lines 506-512, were removed. 

Line 538 – please revise ‘In consistence’ with ‘Consistent with these results’.

Response/Reply:

‘In consistence’ with was changed to ‘Consistent with these results’

Please again try to avoid very short paragraphs as in lines 565-567, 589-591 634-637.

Response/Reply:

Short paragraphs in lines 565-567, 589-591 634-637, were avoided.

A section on the implications of your findings would be better placed prior to the conclusion. Moreover, can the authors offer any insight as to how such factors may influence potential future vaccination decision-making in a post-pandemic landscape? What can we learn from the findings to inform future health promotion efforts? Can the authors also differentiate between factors that influence parental COVID-19 vaccination globally, but also factors specific to Saudi Arabia? 

Response/Reply:

Thanks to the reviewer for this comment.

A section on the implications of this SRMA findings was placed prior to the conclusion. 

Insight as to how such factors may influence potential future vaccination decision-making in a post-pandemic landscape was offered. The differentiating factors that influence parental COVID-19 vaccination globally and factors specific to Saudi Arabia were provided in lines 477-481.

Overall, this was a pleasure to read and review. I commend the authors on their hard work in the production of this high-quality manuscript.

Response/Reply:

Many thanks to the reviewer for his commendation and it was a pleasure and great chance to have such an expert reviewer in the field of this SRMA who could recognize the hard work in the production of this high-quality manuscript.

Reviewer #3: Thank you for the opportunity to review this interesting and well-written paper. I appreciate the effort put into this study. However, I have several concerns and suggestions regarding the methodology and discussion sections:

Response/Reply:

Many thanks to the reviewer for appreciating effort put into this SRMA and acknowledging it as an interesting and well-written paper. 

Line 145: Consider replacing the term "reasons" with "factors" for greater precision.

Response/Reply:

Authors would like to clarify that the term "reasons" is a separate primary objective or outcome of this SRMA and it was used to refer to causes or personal beliefs/opinions of parents who were willing to immunize their children with COVID-19. A qualitative analysis was used to identify and categorize the reasons influencing parental willingness to vaccinate their children against COVID-19 under recurring themes across studies to provide a deeper understanding of the contributing reasons. Table 2. Specifically presented the reasons for parental willingness to immunize their children with COVID-19 vaccines in descending order of frequency. On the other hand, the term "factors" is a secondary or additional question or objective or outcome to identify what factors=determinants=predictors are associated with parents’ willingness to immunize their children with COVID-19 vaccines. For accurate quantitative estimation of the role of these factors, Chi-squared test with calculation of odds ratios (ORs) and 95% confidence intervals (CIs) was used to compare all possibly associated factors between parents who intended and parents who clearly refused to vaccinate their children against COVID-19. Table 3. Presented these determinants (factors/predictors) of parental willingness to immunize their children with COVID-19 vaccines in a descending order of significance.

Line 149: It is unclear why the authors did not include Scopus in the search strategy. Please provide a rationale for this choice.

Response/Reply:

Thanks a lot, to the reviewer for this query.

Scopus has not been included in the search strategy and Web of Science (WOS) and Pubmed were mainly used. Authors would like to provide the rational for this choice. 

First, both Scopus and WOS are respected platforms, and the choice often depends on specific needs, research discipline, and institutional preferences. WOS is preferred in our institute (King Abdulaziz University) which provides free access to WOS database but not for Scopus.

Second, WOS covers fewer journals overall compared to Scopus which has broader journal coverage, including interdisciplinary and regional research. However, Scopus is less restrictive, allowing for the inclusion of regional and emerging journals and aims for wider accessibility and coverage. Many have found the citations covered by Scopus negative and without a peer review. On the other hand, WOS focuses on high-impact journals in core scientific disciplines with a rigorous selection process. WOS focuses on the Impact Factor as the main metric, which is a classic indicator of scientific impact which is widely recognized as a benchmark for journal quality. Thus, WOS can be considered more selective and prestigious for high-quality research and commonly used in academia for high-impact publications. This was of prime importance to include high quality research in this SRMA and considered as a strength point mentioned at lines 687-688 stating that: including studies published in peer-reviewed journals and high-quality studies according to NOS, helped to ensure the quality and reliability of the data of this SRMA

Third, a

---

## [Decision Letter · Decision Letter 1]

9 Jan 2025

Navigating the shots: Parental willingness to immunize their children with COVID-19 vaccines in Saudi Arabia explored through a systematic review and meta-analysis

PONE-D-24-31975R1

Dear Dr. Hegazi,

We’re pleased to inform you that your manuscript has been judged scientifically suitable for publication and will be formally accepted for publication once it meets all outstanding technical requirements.

Kind regards,

Maher Abdelraheim Titi

Academic Editor

PLOS ONE

Reviewers' comments:

Reviewer's Responses to Questions

**Comments to the Author**

1. If the authors have adequately addressed your comments raised in a previous round of review and you feel that this manuscript is now acceptable for publication, you may indicate that here to bypass the “Comments to the Author” section, enter your conflict of interest statement in the “Confidential to Editor” section, and submit your "Accept" recommendation.

Reviewer #2: All comments have been addressed

Reviewer #3: All comments have been addressed

Reviewer #4: All comments have been addressed

2. Is the manuscript technically sound, and do the data support the conclusions?

Reviewer #2: Yes

Reviewer #3: Yes

Reviewer #4: Yes

3. Has the statistical analysis been performed appropriately and rigorously? 

Reviewer #2: Yes

Reviewer #3: Yes

Reviewer #4: Yes

4. Have the authors made all data underlying the findings in their manuscript fully available?

Reviewer #2: Yes

Reviewer #3: Yes

Reviewer #4: Yes

5. Is the manuscript presented in an intelligible fashion and written in standard English?

Reviewer #2: Yes

Reviewer #3: Yes

Reviewer #4: Yes

6. Review Comments to the Author

Reviewer #2: (No Response)

Reviewer #3: (No Response)

Reviewer #4: Thanks for the efforts to address all the comments. The revised version could be accepted for publication.

7. PLOS authors have the option to publish the peer review history of their article (what does this mean?). If published, this will include your full peer review and any attached files.

Reviewer #2: No

Reviewer #3: No

Reviewer #4: No

---

## [Editor Report · Acceptance letter]

15 Jan 2025

PONE-D-24-31975R1 

PLOS ONE

Dear Dr. Hegazi, 

I'm pleased to inform you that your manuscript has been deemed suitable for publication in PLOS ONE. Congratulations! Your manuscript is now being handed over to our production team.

Kind regards, 

on behalf of

Dr. Maher Abdelraheim Titi 

Academic Editor

PLOS ONE
